

Geoscientific
Instrumentation
Methods and
Data Systems

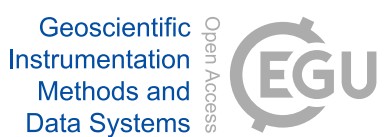

# A compact ocean bottom electromagnetic receiver and seismometer

**Kai Chen[1], Ming Deng[1], Zhongliang Wu[2], Xianhu Luo[2], and Li Zhou[1]**

[1]School of Geophysics and Information Technology, China University of Geosciences (Beijing), Beijing, China
[2]MLR Key Laboratory of Marine Mineral Resources, Guangzhou Marine Geological Survey, Guangzhou, China

**Correspondence:** Kai Chen (ck@cugb.edu.cn)

**Abstract.** Joint marine electromagnetic (EM) and seismic interpretations are widely used for offshore gas hydrate and petroleum exploration to produce improved estimates of lithology and fluids and to decrease the risk of low gas saturation. However, joint data acquisition is not commonly employed. Current marine EM data acquisition depends on an ocean bottom electromagnetic receiver (OBEM), and current seismic exploration methods use seismometers. Joint simultaneous data acquisition can decrease costs and improve efficiency, but conventional independent data receivers have several drawbacks, including a large size, high costs, position errors, and low operational efficiencies. To address these limitations, we developed a compact ocean bottom electromagnetic receiver and seismometer (OBEMS). Based on existing ocean bottom $E$-field TS1 (OBE) receiver specifications, including low noise levels, low power consumption, and low time drift errors, we integrated two induction coils for the magnetic sensor and a three-axis omnidirectional geophone for the seismic sensor to assemble an ultra-short baseline (USBL) transponder as the position sensor, which improved position accuracy and operational efficiency while reducing field data acquisition costs. The resulting OBEMS has a noise level of $0.1\,\mathrm{nV\,m^{-1}\,rt^{-1}}$ TS2 (Hz) at 1 Hz in the $E$-field, $0.1\,\mathrm{pT\,rt^{-1}}$ (Hz) at 1 Hz in the $B$-field, and a 30 d battery lifetime. This device also supports a Wi-Fi interface for the configuration of data acquisition parameters and data download. Offshore acquisition was performed to evaluate the system's field performance during offshore gas hydrate exploration. The OBEMS operated effectively throughout the operation and field testing. Therefore, the OBEMS can function as a low-cost, compact, and highly efficient joint data acquisition method.

## 1 Introduction

Marine electromagnetic (EM) and seismic methods are important geophysical tools used for offshore petroleum exploration (Barsukov and Fainberg, 2017; Constable and Srnka, 2007; Ellingsrud et al., 2002), gas hydrate mapping (Schwalenberg et al., 2017; Weitemeyer et al., 2006), physical oceanography (Zhang et al., 2014), crustal studies (Constable and Heinson, 2004; Key and Constable, 2002; Kodaira et al., 2000), mid-ocean ridge studies (Key, 2012), subduction zone studies (Naif et al., 2015), and underwater target detection (ISL, 2016). To improve the interpretation accuracy and decrease the risks associated with offshore drilling, surveys increasingly conduct multi-physics integrated interpretation and cooperative inversion, such as joint marine EM and seismic interpretations (Goswami et al., 2015, 2017; Weitemeyer et al., 2011). In offshore exploration, seismic and EM data acquisition is typically performed independently. Joint offshore seismic and EM data acquisition would not only increase the efficiency and decrease costs but also improve the interpretation accuracy (Engelmark et al., 2012). Complementary data also enhance our understanding of subsurface characteristics. While seismic methods provide an indication of the subsurface architecture, EM is more sensitive to changes in fluids. Seismic data can be inverted for velocity and acoustic impedance, while EM data inversion provides resistivity values. Correlating these two methods has the potential to improve hydrocarbon saturation estimates and drilling success rates.

Ocean bottom electromagnetic receivers (OBEMs) are used to measure seafloor magnetotelluric (MT) and controlled source electromagnetic (CSEM) field signals. Jean Filloux and colleagues performed the first seafloor EM

measurements made using the MT method in the 1960s (Filloux, 1967). More recently, the Scripps Institution of Oceanography (SIO) collaborated with Quasar to design a small EM receiver (Quasar, 2019), based on the existing SIO EM receiver (Constable, 2013) for compact and low-cost EM data acquisition. The resulting QMax EM3 receiver optimized efficiency and safety, enabling survey contractors to achieve faster deployment and recovery times, use more receivers, and perform more surveys in a shorter time period (Quasar, 2019). Kasaya (2009) also developed a small OBEM and ocean bottom electrometer (OBE) system that had an arm-folding mechanism to facilitate assembly and recovery operations. For magnetic observations, they used a fluxgate sensor. This OBE mainly focuses on marine MT acquisition, and the CSEM is not included. Current trends in instrumentation involve smaller sizes, lower power consumptions, lower noise levels, and lower data acquisition costs. Marine EM field data acquisition technology continues to improve with reduced costs and increased flexibility.

Ocean bottom seismometers (OBSs) have been employed to produce offshore seismographs since the 1970s. OBSs are usually equipped with three component geophones to record sound waves generated by either earthquakes at depth or man-made devices near the surface. They record the movement of the seafloor in all directions, while a hydrophone records the pressure in the surrounding water. The French Research Institute for Exploitation of the Sea (IFREMER) (Auffret et al., 2004) developed a new generation of ocean bottom seismometers by integrating acquisition and instrument release, adding a rechargeable battery, enabling data downloads via USB cable, and reducing the unit's size. Geo-Pro GmbH developed a similar system whose CPU and recorder are housed in a 17 in. TS3 glass sphere. Panahi et al. (2008) designed a low-power data logger for OBS systems based on a compact flash card. Sercel and GeoPro GmbH CE1 are currently the leading manufacturers in the OBS market. All their instruments are designed for low power consumption, low noise levels, low time drift errors, and a compact size.

Marine controlled-source electromagnetic (CSEM) sounding is a new tool available to geophysicists for offshore gas hydrate exploration (Weitemeyer et al., 2011). This technique has been developed to detect deep hydrocarbon reservoirs (Fanavoll et al., 2010). The OBEM is the receiver that measures the EM field for the marine CSEM and/or MT method. The OBS mainly provides deep geological information and is also used for shallow gas hydrate mapping (Mienert et al., 2005). Therefore, these two offshore active and passive geophysical exploration instruments can jointly provide a complementary image to identify natural resources and/or geologic structures. Thus, combining OBEM and OBS data acquisition to investigate gas hydrate or petroleum exploration within a few kilometers below the seafloor is desirable.

Current offshore EM and seismic data acquisition typically employs both OBEM and OBS, but the two instruments operate independently. There are two disadvantages to independent data acquisition: (1) the error related to each individual instrument position and (2) the cost of offshore data acquisition, which includes instrument hardware, research vessels, and human resources, among other factors. Considering the former, the quality of marine CSEM data is dependent on accurate navigational information for both the transmitter and receiver positions and orientations. Current OBEMs locate via acoustic release, such that it has a greater position error and uses the near field to refine the geometry of the transmitter and receiver locations (Weitemeyer et al., 2011), which are dependent on data post-processing. Position errors may lead to reduced inversion accuracy.

Both KMS and GeoSYN have developed a GEOSYN/KMS 870-VectorSeisEM Broad Band-Ocean bottom station CE2, which is a broadband 4C seismic/6C electromagnetic node for shallow and deepwater geophysical applications. Using a single survey vessel, Petroleum Geo-Services, Inc. (PGS) can acquire high-density EM data simultaneously with 2-D GeoStreamer® seismic data or high-density 3-D EM data over existing or planned 3-D seismic data. Offshore high-density joint EM and seismic acquisition and integrated data analysis represent a stepwise change in the application of EM technology. Both technologies seek to mitigate risk when searching for and extracting oil and gas. During 2010, we acquired coincident marine CSEM and OBS data when PGS conducted one of the first field trials of their towed streamer EM system at the Troll field, located in the Norwegian North Sea (Zhdanov et al., 2012). The towed streamer EM system allowed the acquisition of CSEM data simultaneously with seismic data over large areas, resulting in higher production rates and lower costs than those associated with conventional CSEM acquisition.

The China University of Geosciences (Beijing) (CUGB) developed an OBEM in 1998 (Deng et al., 2003). During the past 20 years, CUGB has successfully used its OBEM equipment in deep EM surveys for gas hydrate mapping and hydrocarbon exploration (Wei et al., 2009; Jing et al., 2016). The OBEM has also been widely used for marine magnetotelluric and CSEM measurements. The current OBEM system has an acoustic telemetry modem and a folding-arm mechanism (Chen et al., 2015) with low noise levels and low time drift errors. In 2014, CUGB developed a micro-OBE for low-cost and highly efficient data acquisition (Chen et al., 2017). To achieve joint EM and seismic data acquisition, the instrument was upgraded from an existing micro-OBE by (1) integrating a three-axis omnidirectional geophone for seismic parameter measurements, (2) installing two induction coils for horizontal magnetic field component measurements, and (3) installing an ultra-short baseline (USBL) transponder for tracking the seafloor position as the system ascends after release.

The ocean bottom electromagnetic receiver and seismometer (OBEMS) has been mechanically optimized to sat-

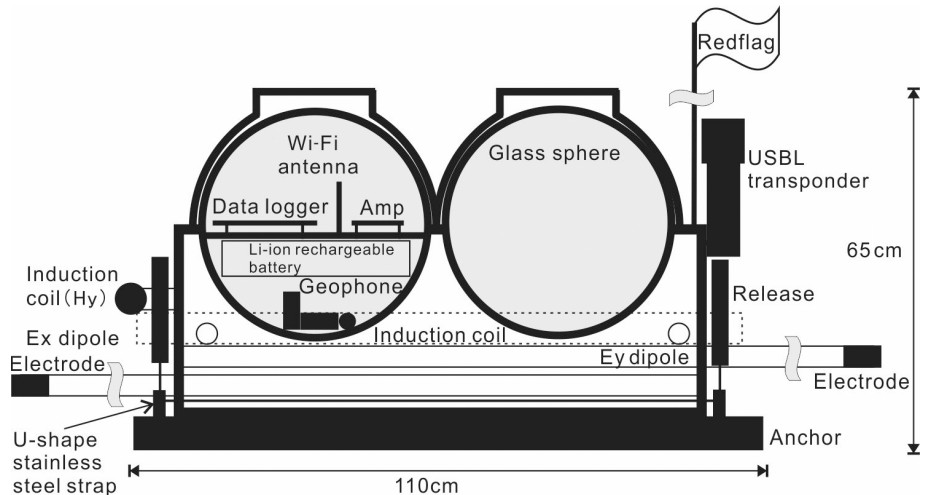

**Figure 1.** Schematic diagram of the ocean bottom electromagnetic receiver and seismometer (OBEMS). Diagram shows the structural design inside the glass sphere, with omnidirectional geophones in the lowest layer followed by the Li-ion rechargeable battery packs, acoustic telemetry modem (ATM), and the data logger. All print circuit boards are covered with magnetic shielding. Ferrite sheets, with a 0.01 mm thick film on one side and a 0.02 mm thick adhesive tape on the other, were glued inside the shielding box. These ferrite sheets function primarily as suppressors, blocking EM noise at lower frequencies and absorbing it at higher frequencies. A single U-shaped stainless steel strap connects the two release mechanisms, passing through two stainless steel loops set into the anchors. TS4

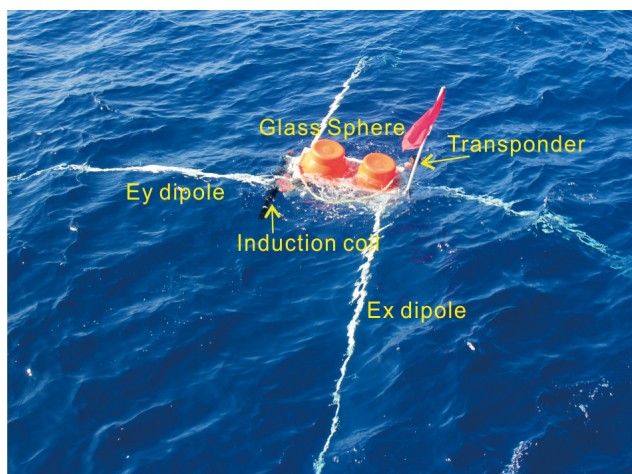

**Figure 2.** Photograph of the OBEMS while in the water ascending to the surface after release. The length of the electrode dipole is 12 m.

isfy all technical requirements for simultaneous joint seismic and electromagnetic data acquisition. This technical advancement permits enhanced modeling and the simultaneous interpretation of both datasets, which minimizes acquisition costs. The advantages of this OBEMS include (1) lower cost and higher efficiency of both the instrument and offshore data acquisition, as the same cost includes more nodes required to improve the horizontal resolution, and (2) a smaller seafloor instrument position error, which decreases the inversion error. The OBEMS system can also replace an OBEM as the receiver in marine CSEM surveys. In addition, the OBEMS can

be used for OBS observations. In the future, a hydrophone will be added to the OBEMS system to allow measurements of the acoustic pressure field.

## 2   Instrument specifications

To achieve joint EM and seismic data acquisition with the objectives of reducing the data acquisition cost and improving operational efficiency, we developed a new OBEMS. The OBEMS was then used to record the seafloor EM field and acoustic signals. Figure 1 shows a schematic of the system. The OBEMS consists of a nylon frame, two glass spheres, a red flag, a transducer, a USBL transponder, a data logger, a battery, three geophones, four electrodes, two induction coils, and an anchor. The equipment is fixed onto a nylon frame measuring $105\,\text{cm} \times 55\,\text{cm} \times 65\,\text{cm}$. All electronics are installed inside a 17 in. TS6 glass sphere, except for the transducer and USBL, while the other glass sphere provides buoyancy. Figure 2 shows a photo of the OBEMS while floating to the surface. We used Ag/AgCl electrodes to measure the electric voltage in the $E_x$ and $E_y$ TS7 dipoles. The $E$-field noise level was $0.1\,\text{nV}\,\text{m}^{-1}\,\text{rt}^{-1}$ (Hz) at 1 Hz (at a working water depth of 1000 m) with a 12 m dipole. We used commercial omnidirectional geophones (ODG8 geophone, manufactured by Chongqing Geological Instrument Factory) as seismometers, with an 8 Hz natural frequency, to record three artificial orthogonal earthquake signal components. Omnidirectional geophones were used because traditional geophones cannot effectively and reliably receive vibration signals on an inclined seabed. In addition, an atti-

**Table 1.** Technical specifications for the ocean bottom electromagnetic receiver and seismometer (OBEMS).

| | |
|---|---|
| Channels | Seven: $E_x/E_y$, geophone $(x, y, z)$, $H_x$, $H_y$ |
| Sensor type | Electrode: Ag/AgCl<br>Magnetic sensor: induction coil<br>Geophone: triaxial, orthogonal, omnidirectional moving coil |
| Channel $-3$ dB bandwidth at fs $= 2400$ Hz TS5 | $E$-field: 0.01 to 100 Hz<br>Induction coil: 0.1–500 Hz<br>Geophone: 8–300 Hz |
| Maximum battery lifetime | 30 d |
| Memory | 32 GB SD card (upgrade 128 GB) |
| Release mechanism | USBL motor drive release and burn wire release |
| Weight | 114 kg in air (exclude anchor)<br>$-32$ kg in water |
| Electrode dipole | 12 m |
| Dynamic range | $E$-field channel: 110 dB; $B$-field channel: ~~130~~ dB (both at fs $=$ ~~1000~~ Hz CE3)<br>Seismic channel: ~~130~~ dB at fs $=$ ~~1000~~ Hz |
| Position in water | USBL responder |
| Sensitivity | Geophone: 78.5 V m$^{-1}$ s$^{-1}$ at 15 Hz<br>Induction coil: 0.3 V nT$^{-1}$ |
| Gain preamplifier | $E$-field channel: 480 to 30 720 step 2<br>$B$-field channel: 0.4 to 25.6 step 2<br>Geophone: 20 to 56 dB step 6 dB |
| Noise level | $E$-field: 0.1 nV m$^{-1}$ rt$^{-1}$ (Hz) at 1 Hz<br>$B$-field: 0.1 pT rt$^{-1}$ (Hz) at 1 Hz |
| Time drift error | Less than 2 ms d$^{-1}$ |
| Maxim work water depth | 4000 m |
| User interface | USB & Wi-Fi (data transfer rate 3 MB s$^{-1}$) |

tude and heading reference system (AHRS) was installed to measure the orientation and inclination of the geophone for further data processing. The moving coil geophone may generate EM noise on the magnetic sensors, but the electronics (data acquisition circuit board, battery, and geophone) are all shielded by ferrite film, and the distance between the induction coil and geophone is too large to measure the EM noise. We confirmed the EM noise of the geophone by testing it in a magnetically shielded room. The geophone sensitivity was 78.5 V m$^{-1}$ s$^{-1}$ at 15 Hz, and the internal resistance was 3100 Ω. Four sets of six 18 650 Li-ion rechargeable batteries (for 25.2 V) supply power to the data logger circuit. One independent 16.8 V battery supplies power to the acoustic telemetry modem (ATM) module. The power consumption is approximately 1 W at a maximum sampling rate of 2400 Hz, and the power supply module supports data acquisition for $\sim 30$ d.

The OBEMS data logger has a 24 bit analog-to-digital converter for each of the two electrical field components and the three-axis geophone components. The attitude and heading reference system (AHRS) module records the pitch, roll, and heading while the instrument is on the seafloor. The OBEMS has two parallel release mechanisms. The transducer connects with the ATM for acoustic telemetry. When the ATM receives the release command, the burn wire mechanism release is triggered CE4, and the anchor releases after 10 min. Additionally, a USBL transponder responder and motor-driven release were installed on the OBEMS. The transponder is designed for positioning remotely operated vehicles (ROVs), towed fish, and other mobile targets in water depths up to 4000 m and is equipped with an omnidirectional transducer for a wide range of general USBL tracking applications. The transponder is available with acoustically controlled output lines suitable for an external motor drive. This transponder is integrated with the USBL transponder,

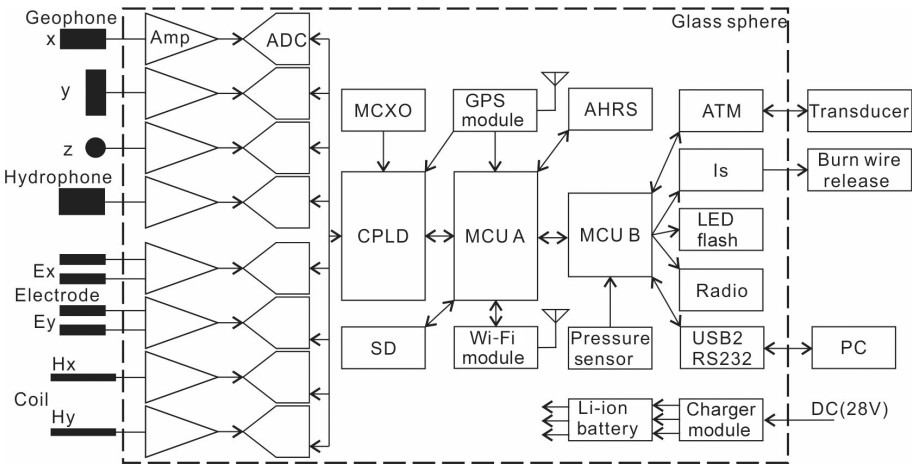

**Figure 3.** Circuit diagram of the OBEMS data logger, containing EM and seismic sensors, eight amplifiers, eight analog-to-digital converters (ADC), a complex programmable logic device (CPLD), secure digital memory card (SD), microprocessor controlled crystal oscillator (MCXO), Li-ion rechargeable battery, charger module, LED flash, radio, ATM, and attitude and heading reference system (AHRS). In this preliminary design, the ATM, burn wire release, ultra-short base line (USBL), and relay release are all designed for release.

release, and an internal depth sensor to improve USBL position performance.

The resulting reduction in the positioning uncertainty leads to significant improvements in the target sensitivity. Acoustic ultra-short baseline communication (USBL) is used to establish the exact receiver positions. The OBEMS integrated a USBL transponder from Sonardyne (Gyro USBL) to determine the underwater acoustic positioning, whose accuracy is approximately 1.5‰ of the slant distance. When the slant distance is 2000 m, we estimate that the receiver positions obtained in this manner are accurate to approximately 3 m. The position of the OBEM from the EMGS is monitored by the acoustic USBL transponders. The accurate navigational data from the SIO OBEM were collected using a short baseline (SBL) acoustic navigation system. The receiver positions obtained in this manner are accurate to approximately 3–5 m. The USBL is more convenient to install and use than the SBL, and it has sufficient accuracy.

To maintain a simple and compact OBEMS design, the seismometers were indirectly coupled to the seafloor via the sphere, release hardware, anchor, and spring used to connect the OBS to the anchor. While seismometers work best when they are in direct contact with the Earth (Mànuel et al., 2012), this design has been proven effective for collecting data at long shot–receiver offsets. Coupling the instrument to the seafloor is important, as the geophone, which measures seafloor movement, is located inside the sphere rather than deployed on the seafloor. To further optimize coupling, a weight in the shape of a cross with a U-shaped strap was used to ensure good penetration of the anchor weight into the seafloor. The single U-shaped stainless steel strap connects the two burn wire mechanisms, passing through two stainless steel loops set into the anchors.

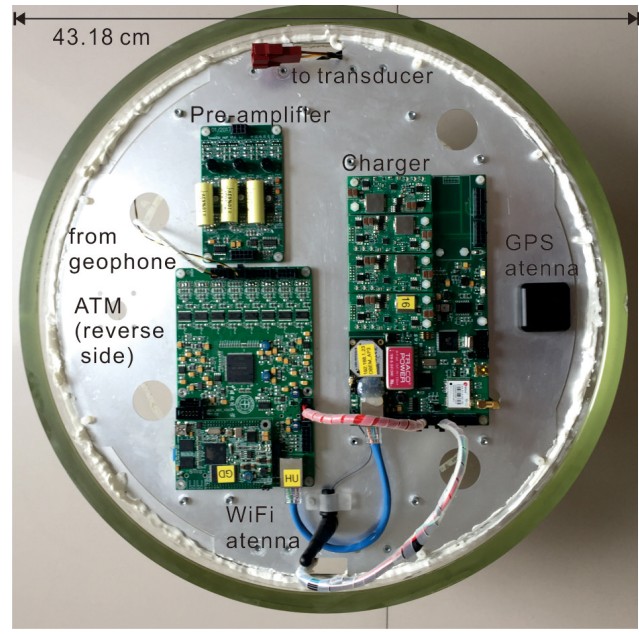

**Figure 4.** Photo of the data logger installed in the glass sphere. Under the aluminum board, an ATM, Li-ion rechargeable battery set, and three-axis geophone are fixed. After circuit assembly, magnetic shielding covers the three print circuit boards to decrease sensor disturbance.

To provide the highest possible accuracy, time was recorded to the nearest millisecond. Each OBEMS uses a microprocessor controlled crystal oscillator (MCXO) as a stable clock reference, for which the drift can be as little as $2\,\mathrm{ms}\,\mathrm{d}^{-1}$. Following each deployment, the offset was measured to compensate for the total time drift.

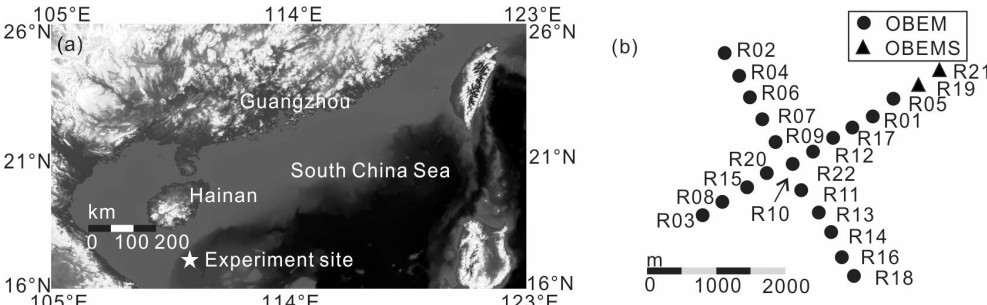

**Figure 5.** Photograph of the offshore experiment area and site map with all receiver locations marked. The map was generated using the Global Mapper software package.

The size of the anchor is $110\,\text{cm} \times 60\,\text{cm} \times 6\,\text{cm}$, weighing $136\,\text{kg}$ in air and $78\,\text{kg}$ in water. The weight of the OBEMS in water is $42\,\text{kg}$ during deployment and $-36\,\text{kg}$ upon ascent. The redundant buoyancy is designed for the addition of more batteries for a longer seafloor working time. The descent and ascent velocities of the OBEMS are approximately 1 and $0.8\,\text{m}\,\text{s}^{-1}$, respectively. Table 1 lists the specifications of the OBEMS system.

## 3 Electronics

Figure 3 shows a schematic of the OBEMS data logger. The data logger is based on a 24 bit analog-to-digital converter (ADC) for each channel. Different analog preamplifiers were used for the electric field, magnetic field, and geophone measurements. The data logger contains eight channels. Each channel integrates a preamplifier and a 24 bit ADC. The preamplifier for the $E$-field channel is an ultra-low-noise chopper amplifier that has been upgraded from Constable (2013). The self-noise level is approximately $0.6\,\text{nV}\,\text{rt}^{-1}$ (Hz) at $1\,\text{Hz}$, the gain is 1200, and the $-3\,\text{dB}$ bandwidth is 0.01 to $100\,\text{Hz}$. The preamplifier for the geophone is a different amplifier with a gain of $20\,\text{dB}$. The gain of the induction coil is $300\,\text{mV}\,\text{nT}^{-1}$, and the output range is $\pm 5\,\text{V}$. The input range of the ADC is $\pm 5\,\text{Vpp}$ to match the preamplifier and ranges of the induction coil output. The ADC module is based on an eight-channel, 24 bit ADC, i.e., the ADS1282 (Texas Instruments). The ADS1282 is a one-channel, high-dynamic-range, fourth-order $\Delta$-$\Sigma$ modulator, with a digital filter for data decimation and interfacing with the microcontroller module, which provides a dynamic range of $130\,\text{dB}$ at a $250\,\text{Hz}$ sampling rate and a total harmonic distortion (THD) of $-122\,\text{dB}$. The full scale of the ADC is $\pm 2\,\text{V}$ with an attenuation coefficient of 0.4.

A microcontroller unit (MCU) A (AT91SAM9G45 from Atmel) is the master MCU, which is used to do the following: set the sampling rate; configure the ADC register; write data to the SD card; communicate with a computer via a Wi-Fi module; and communicate with the slave MCU B (ATmega16 from Atmel), the attitude and heading reference system (AHRS) module, and the GPS module via a serial port. The complex programmable logic device (CPLD) (EPM570 from Intel) reads, in parallel, converted data from the eight ADCs and series data awaiting MCU A data transfer. The MCU A employs an internal direct memory access (DMA) controller and writes data to the SD card. The MCXO (MX-503 from Vectron) has a low power consumption of $3.3\,\text{V}$ and $12\,\text{mA}$, with a high-frequency stability of approximately $\pm 20\,\text{ppb}$ from 0 to $50\,^\circ\text{C}$. The CPLD generates a $2.4576\,\text{MHz}$ clock as the ADC master clock. The selectable sampling rate can be set to 2400, 600, or $150\,\text{Hz}$, with a dynamic range that reaches approximately 115, 121, and $127\,\text{dB}$, respectively.

MCU B is used as the slave MCU for communication with the ATM, driving the burn wire current source, measuring battery capacity, and acting as a pressure sensor inside the glass sphere for negative pressure monitoring to determine if there is a leak in the glass sphere. When the ATM receives the release command, MCU B triggers the current source and provides $500\,\text{mA}$ to the burn wire release. A charger module

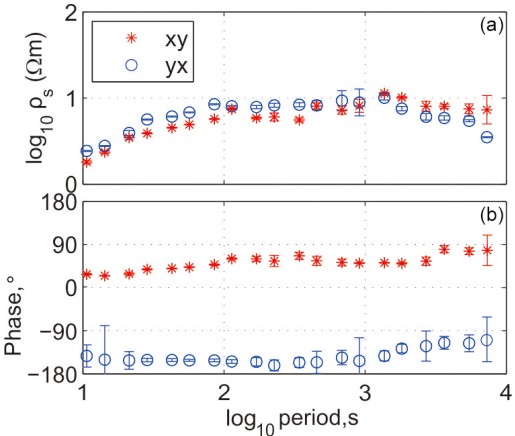

**Figure 6.** Apparent resistivity **(a)** and phase **(b)** curves calculated from the OBEMS $E$-field and $B$-field at site R19. The red stars indicate the $xy$ components, and the blue circles indicate the $yx$ components.

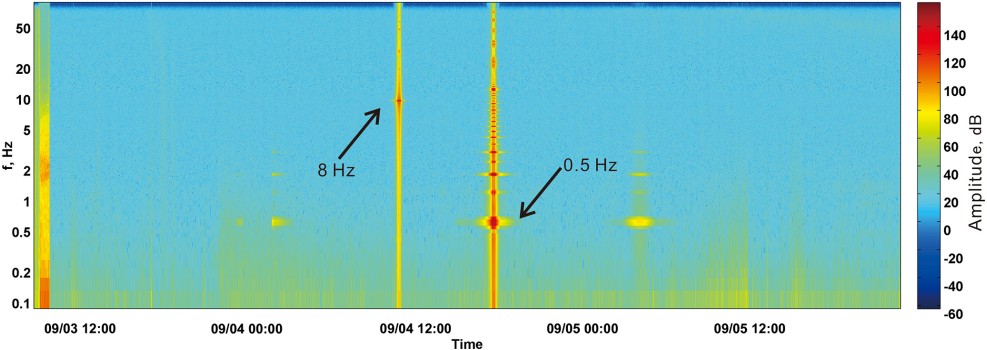

**Figure 7.** Short-time Fourier transform results for the CSEM signal from the horizontal electrical field components.

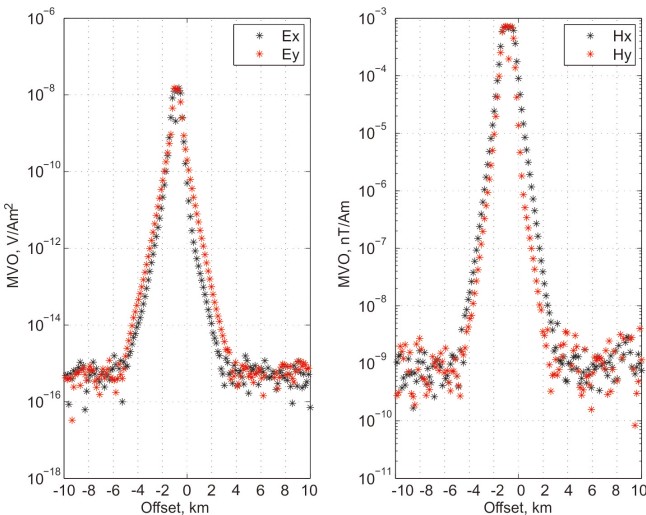

**Figure 8.** Magnitude versus offset for the *E* and *H* components (site: R19; frequency: 8 Hz).

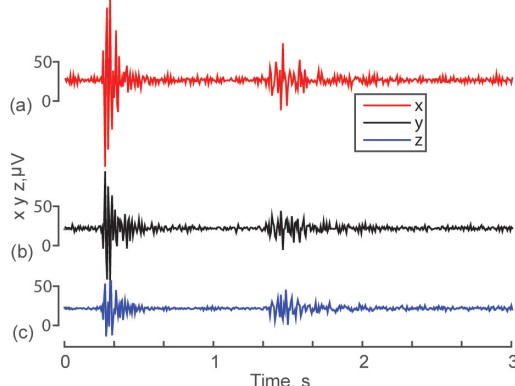

**Figure 9.** Seismic data acquisition test results in real environmental conditions from the OBEMS at test site R21. **(a)** First horizontal component, **(b)** second horizontal component, and **(c)** vertical geophone component.

converts the external DC 28 V to independently charge each Li-ion rechargeable battery pack.

The collected data are stored on an SD card. To download the data, there is no need to open the glass sphere or use an Ethernet cable. A computer can connect to the data logger to download the data and configure acquisition parameters using the onboard Wi-Fi module. The capacity of the SD card is 32 GB, which can be expanded to 128 GB. The sampling rate was set to 150 Hz, generating 400 MB of data per day. An effective download speed of 3 MB s$^{-1}$ was achieved, which allows 30 d of data (approximately 12 GB) to be downloaded in less than 67 min. After the OBEMS is released from its anchor and floats to the surface, it can be recovered using radio signals detected by a 165 MHz very-high-frequency (VHF) direction finder at distances of up to 5 km even in poor visibility. The flashing light inside the sphere is especially useful for recovery at night. Figure 4 shows a photo of the data logger installed in the glass sphere.

## 4 Offshore experiments

In August 2018, we conducted offshore experiments to map gas hydrates in the Qiongdongnan region of the South China Sea. The support vessel used was the *Hai Yang Si Hao*, which is registered with the Guangzhou Marine Geological Survey Bureau. This expedition was the result of a collaboration between the Guangzhou Marine Geological Survey and CUGB. The scientific objective of the cruise was to map gas hydrates using a marine EM method while also using CSEM to determine the electrical structure 500 m below the seafloor. To achieve this, 20 previously developed OBEMs and a towed CSEM transmitter (Wang et al., 2017) were operated during the cruise. To evaluate the overall performance of the receiver described in this study, two OBEMS were also used.

The Qiongdongnan region of the South China Sea is located 170 km southeast of Sanya. The seafloor is an ocean basin with depths of 1700–1800 m. Figure 5 shows a map of the experimental area, which included 22 receivers. Here, R19 and R21 indicate the two newly developed OBEMS, while other labels represent the other existing OBEMs. All

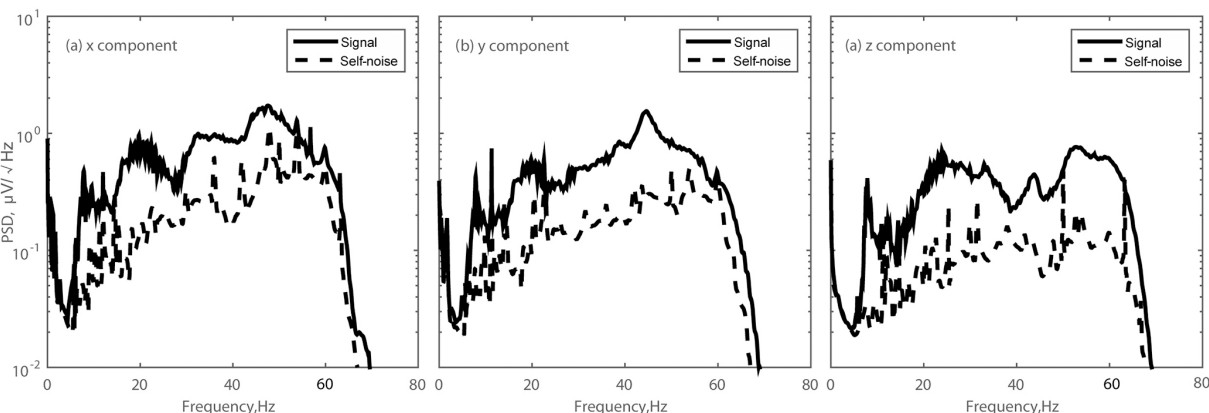

**Figure 10.** Power spectral density (PSD) of the signal and self-noise in real environmental conditions from the OBEMS at test site R21. **(a)** First horizontal component, **(b)** second horizontal component, and **(c)** vertical geophone component.

receivers were equipped with USBL transponders. When the receiver began its descent to the seafloor, the transponder tracked its position. If the depth of the receiver did not change substantially, the location result was assumed to indicate its landing position. After all receivers were deployed, two transmitter lines were towed for different waveforms: a single square waveform at 8 Hz and multiple frequencies synthesized with a 0.5 Hz fundamental frequency. After 10 d on the seafloor, all 20 OBEMs and the 2 OBEMS units were successfully recovered.

We estimated the MT responses (apparent resistivities and phase differences) using the robust estimate method (Egbert, 1997). To calculate the MT responses at site R19, Fig. 6 shows the respective computed MT responses for the site over a period of 10 to 10 000 s. The data quality was excellent down to periods of approximately 10 000 s. At high frequencies, we observed the sea floor response for both modes asymptotic to 1 Ωm.

The CSEM data acquisition employed a towed CSEM transmitter with a length of 300 m that generated horizontal electrical dipoles (HEDs), while the altimeter of the towed transmitter body was approximately 20–50 m. The transmitters were used at a single frequency of 8 Hz and multiple frequencies synthesized with a fundamental frequency of 0.5 Hz. The transmitter was equipped with a depth sensor, altimeter, and acoustic transponder. The transmitter transmitted at 450 A. The OBEMS received CSEM data, followed by the performance of the horizontal $E$-field- and $B$-field-component fast Fourier transforms (FFTs), current data FFT, instrument calibration, and field component rotation. Figure 7 shows the $E_x$ component of site R19's signal spectrum. The result of the short-time Fourier transform (STFT) clearly shows the two towed CSEM lines (i.e., the 8 and 0.5 Hz towed lines). Figure 8 shows the MVO of the horizontal EM component at site R19. The 8 Hz data are above the instrumental noise floor at a 3.5 km range.

After the CSEM data acquisition using a towed transmitter source, seismic data acquisitions were performed by testing the functionality of the OBEMS using an air gun as the source. Figure 9 shows the single shot waveforms. The vertical peaks in each channel correspond to the acquisition of the reflected and refracted acoustic signals generated by the artificial source. The recordings show clear vibration signal arrivals, which demonstrate the proper functioning of all three geophone channels. Figure 10 shows the power spectra of the three components estimated using the 200 s time series and the ground noise power spectra. The main air gun energy was focused at 40–60 Hz. Due to the low sampling rate (150 Hz), higher-frequency band signals are depressed.

## 5 Conclusions

To achieve joint marine EM and seismic data acquisition, we developed an OBEMS based on an existing micro-OBE receiver, which consisted of two induction coils for horizontal magnetic field component measurements and a three-axis omnidirectional geophone for recording seafloor movement in all directions, as well as an assembled USBL transponder for seafloor position tracking. The final system included four electrodes, three geophones, two induction coils, two glass spheres, a USBL transponder, a motor drive release, an integrated ATM, a burn wire release mechanism, a recovery beacon (LED, radio modem, and VHF radio), and an expanded Wi-Fi module for data transfer. A 17 in. TS8 glass sphere contains the data logger and battery. The proposed OBEMS architecture exhibited low noise, low time drift, and low power specifications. The OBEMS was mechanically optimized to satisfy all technical requirements for the simultaneous acquisition of seismic and electromagnetic data. However, the following minor technical improvements will be made in future studies:

Please note the remarks at the end of the manuscript.

1. The autonomy of the instrument will be extended to 60 d of data acquisition and 90 d on the seafloor.

2. A hydrophone will be installed to achieve a fully integrated all-in-one receiver.

3. Enhancements will be made to the Wi-Fi module performance.

As these preliminary tests have shown, OBEMS technology is capable of high-quality MT, CSEM, and artificial seismic data acquisition. Future developments to this instrument will add a hydrophone and lengthen the seafloor working time (2–3 months), which add to the existing advantages of the OBEMS (i.e., low cost, easy deployment, small size, and high efficiency). These developments will occur through cooperation between the GMGS and CUGB.

*Data availability.* The raw data from the experiments are available upon request (ck@cugb.edu.cn).

*Author contributions.* KC designed the new instrument and wrote the paper. MD was the project leader. ZW and XL provided ideas and guidance, including on the experiment. LZ designed and tested the electronics.

*Competing interests.* The authors declare that they have no conflict of interest.

*Acknowledgements.* The development of the OBEMS was supported by the National High Technology Research and Development Program of China (nos. 2016YFC0303100 and 2017YFF0105700) and as a key project by the National Science Foundation of China (nos. 61531001 and 41804071). The development of the low-noise electrode and amplifier was supported by the Fundamental Research Funds for the Central Universities (nos. 2652015403 and 2652018265). Joint data acquisition was partially supported by the China Geological Survey (no. GZH-201100307). We also acknowledge the extensive support offered by the captain, ship's crew, and marine technicians of the R/V *Hai Yang Si Hao*, as well as Jing TS9 for EM data processing, Zhao TS10 for technical help, and Tu TS11 for helpful comments and support. The instruments were carefully manufactured and modified by Zhang TS12.

*Financial support.* This research has been supported by the National High Technology Research and Development Program of China (grant nos. 2016YFC0303100 and 2017YFF0105700), the National Natural CE6 Science Foundation of China (grant nos. 41804071 and 61531001), and the Fundamental Research Funds for the Central Universities (grant nos. 2652015403 and 2652018265). TS13

*Review statement.* This paper was edited by Francesco Soldovieri and reviewed by two anonymous referees.

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

## Remarks from the language copy-editor

## Remarks from the typesetter