# Peer review of "A Compact Ocean Bottom Electromagnetic Receiver and Seismometer"

_Geoscientific Instrumentation, Methods and Data Systems, 2019_

## Referee Comment (RC1) · Anonymous Referee #1 · 28 Oct 2019

The manuscript has integrated OBEM and OBS into a compact OBEMS system used for offshore gas hydrate and petroleum exploration. The OBEMS system could probably improve the efficiency in fieldwork, but the scientific purpose is uncertain. In general, the CSEM is a useful tool for mapping gas hydrate whereas multiple channels of reflection seismic exploration used in petroleum exploration. These two offshore active geophysical explorations have been jointly used to provide a complementary image to identify natural resources and/or geology structure. The target depth is less than a few kilometers. OBS mainly provides deep geological information extracted by the refracted wave in which the lateral resolution is less than reflection seismic exploration, whereas OBEM has provided a deep sounding. It seems that the instrument has only installed a seismometer into the OBE (Chen et al., 2017) and the OBEM platform

(Chen et al., 2015). Thus, I don't really understand how to join the OBEM and OBS data to investigate gas hydrate or petroleum exploration within a few kilometers below seafloor? I would recommend the authors to distinguish what is the scientific purpose of the instrument? Although the authors have claimed that the seismic signal came from the air gun source that is an insufficient demonstration, more evaluations related to the seismometer and the signal should be required. How to avoid the seismometer generates noise for magnetic sensors? Finally, my personal think that the manuscript should be rejected.

Minor comments: 1. Please comparing and demonstrating the accuracy between the USBL attached to the OBEMS and other OBEM. 2. I can't find the related descriptions of figures 2 and 4 in the context. Please either add the descriptions or remove these figures. 3. P5, L159: How about the gain of the magnetic sensors? 4. P6, L206: Which method? Please cite the reference or specify it in detail. 5. P6, L209: At high frequency ranges, the seafloor responses......Please rewrite it. 6. P6, L198: Figure 7 should be replaced by figure 5? 7. P6, L218: Fig.8 should be replaced by Fig. 7? 8. P6, L220: Figure 9 should be replaced by Figure 8? 9. P6, L224: Fig.10 should be replaced by Figure 9? 10. Table 1 should specify the seismometer in detailed.

---

## Short Comment (SC1) · 5 Nov 2019

A Compact Ocean Bottom Electromagnetic Receiver and Seismometer (ID: gi-2019-25) Response to Reviewers

Dear Anonymous Referee,

Thank you very much for your comments and suggestions. Thank you for your positive comments on this manuscript. According to your advice, we have revised this manuscript as follows.

Comment #1: Thus, I don't really understand how to join the OBEM and OBS data to investigate gas hydrate or petroleum exploration within a few kilometers below seafloor? I would recommend the authors to distinguish what is the scientific purpose of the instru-

[Figure]

ment? Response #2: Marine controlled-source electromagnetic (CSEM) sounding is a new tool available to geophysicists for offshore gas hydrate exploration (Weitemeyer, 2011). And the technique has been developed for the detection of deep hydrocarbon reservoirs (Fanavoll, 2010). The OBEM is the receiver which measure the EM field for the marine CSEM or/and MT method. OBS mainly provides deep geological information, and it also used to shallow gas hydrate mapping (Mienert, 2005). Therefore, these two offshore active/passive geophysical explorations instrument could jointly provide a complementary image to identify natural resources and/or geology structure. Thus, join the OBEM and OBS data acquisition to investigate gas hydrate or petroleum exploration within a few kilometers below seafloor is available. Comment #3: How to avoid the seismometer generates noise for magnetic sensors? Response #3: Three 8 Hz omni-directional geophones were used as seismometer to record artificial earthquakes signal. The moving coil geophone may generate EM noise for magnetic sensors, but the electronics (data acquisition circuit board, battery and geophone) are all shield by ferrite film, and the distance between induction coil and geophone is too large to measure the EM noise. We confirm the EM noise of geophone test in magnetic shield room. Comment #4: Please comparing and demonstrating the accuracy between the USBL attached to the OBEMS and other OBEM. Response #4: The resulting reduction in positioning uncertainty leads to significant improvements in target sensitivity. Acoustic ultra-short baseline communication (USBL) is used to establish the exact receiver positions. The OBEMS integrated USBL transponder which is from Sonardyne GyroUSBL underwater acoustic positioning solution, and the accuracy is approximately 1.5 ‰ of the slant distance. While the slant distance is 2000m, we estimate that receiver positions obtained this way are accurate to about 3 m. The OBEM which is from EMGS position is monitored by acoustic USBL transponders. The OBEM which are from SIO accurate navigational data were meant to be collected using a short baseline (SBL) acoustic navigation system. They estimate that receiver positions obtained this way are accurate to about 3-5 m. The USBL is more convenient to install and use than SBL, and the accuracy is enough. Comment #5: I can't find the related descriptions of figures 2 and 4 in the context. Please either add the descriptions or remove these figures. Response #5: Related descriptions of figures 2 and 4 have been added in the context. Figure 2 show the photo of the OBEMS while floating up on sea level. Figure 4 shows the Photo of the data logger installed in the glass sphere. Comment #6: P5, L159: How about the gain of the magnetic sensors? Response #6: The gain of magnetic sensor is 300mV/nT, and the output range is ±5V. Comment #7: P6, L206: Which method? Please cite the reference or specify it in detail. Response #7: The reference has been added. (Egbert, G. D., Robust multiple-station magnetotelluric data processing, Geophys. J. Int., 130, 475– 496, 1997.) Comment #8: P6, L209: At high frequency ranges, the seafloor responses. Please rewrite it. Response #8: At high frequencies we see the sea floor response for both modes asymptote to 1Ωm. Comment #9: P6, L198: Figure 7 should be replaced by figure 5? P6, L218: Fig.8 should be replaced by Fig. 7? P6, L220: Figure 9 should be replaced by Figure 8? P6, L224: Fig.10 should be replaced by Figure 9? Response #9: We are very sorry for this carelessness. Figure 7 has been replaced by figure 5. Figure 8 has been replaced by figure 7. Figure 9 has been replaced by Figure 8. Figure 10 has been replaced by Figure 9. Comment #10: Table 1 should specify the seismometer in detailed. Response #10: Sensor type, dynamic range and gain preamplifier of the seismometer have been added in table 1.

Once again, thank you very much for your comments and suggestions. We tried our best to improve the manuscript and we have made all of the necessary changes in the manuscript. We truly appreciate the time and efforts of the editors and reviewers, and we sincerely hope that our corrections will meet your approval.

Sincerely, Kai Chen

China University of Geosciences Beijing 100083 ck@cugb.edu.cn

Please also note the supplement to this comment: https://www.geosci-instrum-method-data-syst-discuss.net/gi-2019-25/gi-2019-25-SC1-supplement.pdf

**Supplement:**

**A Compact Ocean Bottom Electromagnetic Receiver and Seismometer**

Kai Chen[1], Ming Deng[1], Zhongliang Wu[2], Xianhu Luo[2], Li Zhou[1]

[revised manuscript text omitted]

---

## Referee Comment (RC2) · Anonymous Referee #1 · 6 Nov 2019

The figure 4 shows a 70.8 km long high-resolution seismic reflection profile JM00-026 crossing both the Storegga (Mienert et al., 2005) instead of OBS data! As I mentioned the critical key point is that the OBS data can't provide a high-resolution profile as seismic reflection profile. Otherwise, could the authors demonstrate how OBS provides a BSR indicator for gas hydrate?
* * *

---

## Referee Comment (RC3) · Anonymous Referee #1 · 6 Nov 2019

Seismic reflection data were acquired in 2000 onboard R/V Jan Mayen, using a double sleeve gun (0.6 l each) towed at w4 m depth and a floating single-channel streamer at short offsets. The frequency content ranges from 30 to 450 Hz, with peak frequency centred around 100 Hz. The overall quality of the records is high, with good signal-tonoise ratio and a sub-surface penetration of up to 1 s. Seismic profile JM00-026 presented in this paper (Fig. 4) is 71 km long, with an average shot spacing of 27.2 m. Data processing included frequency filtering, amplitude corrections and Stolt migration (Mienert et al., 2005).
* * *

---

## Short Comment (SC2) · 15 Jan 2020

Due to the small number and sparseness of the OBS, as well as the limited extent of the shot patch, it would be unrealistic to expect an image of comparable quality to the towed-streamer image. However, OBS is feasible and may be a powerful technology for deep water imaging projects. The superior quality seen in the OBS data is attributed to the following (Manley et al, 2005): 1) Geophone deployment in a quiet, seafloor environment, allowing good coupling with the seafloor may improve the signal-to-noise rate. 2) Wide-azimuth acquisition, enabling discrimination against overburden features that can potentially scatter energy. 3) High fold—the fold of the PZ data is up to 10 times that of the streamer data, enabling a boost in signal-to-noise. 4) PZ summation, enabling receiver-side water-bottom multiple attenuation.

[Figure]

Besides, the CSEM is a new tool available to geophysicists for offshore gas hydrate exploration. Therefore, the OBEMS can used to active EM and seismic method for shallow gas hydrate mapping instead of replacing towed-streamer method.

References: Manley, Dominic & Mohammed, Sean & Robinson, Nigel & Thomas, Rowland. (2005). Structural interpretation of the deepwater Gunashli Field, facilitated by 4-C OBS seismic data. The Leading Edge. 24. 922-926. 10.1190/1.2056396.

---

## Referee Comment (RC4) · Anonymous Referee #2 · 2 Mar 2020

[referee-annotated manuscript omitted]

---

## Author Comment (AC1) · 16 Mar 2020

A Compact Ocean Bottom Electromagnetic Receiver and Seismometer (ID: gi-2019-25) Response to Reviewers

Dear Anonymous Referee,

Thank you very much for your comments and suggestions regarding this manuscript. The comments were very valuable and helpful for revising and improving our manuscript, as well as for elucidating the significance of our research. We have studied all the comments carefully and have made the necessary corrections, which we hope meet your approval. All changes have been indicated using red color. According to your advice, we have revised this manuscript as follows.

[Figure]

Comment 1: Each single shot waveform, its spectrum, and the background noise spectrum should be indicated in the figure. Response 1: Single shot waveforms, spectra, and background noise spectra have been added to Figures 9 and 10 in the revised manuscript. Comment 2: Omni direction geophone without the leveling (gimbal).Because of the ocean bottom instrument has some amount of tilt on the seafloor, 3-comp. geophones should not be true UD/H1/H2 output. The tilt may be estimated by cross correlations between channels. Response 2: Omnidirectional geophones were used because traditional geophones cannot effectively and reliably receive vibration signals on an inclined seabed. In addition, an AHRS was installed to measure the orientation and inclination of the geophone for further data processing. Comment 3: Problem of the use of the flag is a high level noise source at the seafloor, which is well known in the OBS world. Response 3: This is an unfortunate deficit in instrumental design. We will remove the flag in future deployments. Comment 4: reference in Line 49 and Line 51 Response 4: "QUASAR, 2016" has been corrected to "QUASAR 2019" in the revised manuscript. Comment 5: Difference between "towed streamer EM system" and "towed-streamer EM system" from line 86 to line 88. Response 5: "towed-streamer EM system" has been corrected to "towed streamer EM system" in the revised manuscript. Comment 6: First appearance of the "OBEMS", in line 101. Response 6: "Ocean Bottom Electromagnetic Receiver and Seismometer (OBEMS)" has been added to the text. Comment 7: Indicate the model name/number and company of the omni-directional geophone in line 119. Response 7: We have added this information to the revised manuscript. The instrument was an ODG8 geophone manufactured by the Chongqing Geological Instrument Factory. Comment 8: "Li-ion batteries" in line 121 should be corrected as "Li-ion rechargeable batteries" Response 8: "Li-ion batteries" has been corrected to "Li-ion rechargeable batteries" in the revised manuscript. Comment 9: Please add the condition of this power consumption clearly in line 124. Response 9: The power consumption was approximately 1 W at a maximum sampling rate of 2400 Hz and the power supply module supported data acquisition for 30 days. We have added this information to the revised manuscript. Comment 10:

No explanation about the "U-profile". Please add some text here or in Fig.1. in line 144. Response 10: A single U-shaped stainless steel strap connects the two release mechanisms, passing through two stainless steel loops set into the anchors. An explanation of this "U-shape" has been added to the text and to Fig. 1 in the revised manuscript. Comment 11: MCXO should be explain as "microprocessor controlled crystal oscillator" in line 147. Response 11: MCXO has been defined as a "microprocessor controlled crystal oscillator" in the revised manuscript. Comment 12: This buoyancy looks large, please add the reason why it is required in line 151. Response 12: The redundant buoyancy is designed for adding more batteries for longer seafloor working times. We have added this explanation to the text. Comment 13: "24-bit ADC" should be corrected as "a 24-bit ADC" in line 158. Response 13: "24-bit ADC" has been corrected to "a 24-bit ADC" in the revised manuscript. Comment 14: Chopper should be an OP-amp, please indicate model number and company from line 158 to line 159. Response 14: The pre-amplifier for the E-field channel is an ultra-low noise chopper amplifier that has been upgraded from Constable (2013). We have added this information to the text. Comment 15: "ADS1282" should be corrected as "a ADS1282 (TI)" in line 162. Response 15: "ADS1282" has been corrected to "ADS1282 (TI)" in the revised manuscript. Comment 16: This value is in your OBEMS system or the ADC only? Please indicate clearly in line 165. Response 16: This is for ADC only. We have added this information to the text. Comment 17: Indicate model and company of the MCU in line 166. Response 17: This is an ATmega16 from ATMEL. We have added this information to the text. Comment 18: Indicate model and company of the CPLD in line 169. Response 18: This is an EPM570 from INTEL. We have added this information to the text. Comment 19: Indicate the model of the MCXO inline 171. Response 19: This is a MX-503 from Vectron. We have added this information to the text. Comment 20: The resolution in each sampling rate should be different, please indicate from line 173 to line 174 Response 20: The sampling rate can be set to 2400 Hz, 600 Hz, or 150 Hz, and the dynamic range reaches approximately 115 dB, 121 dB, and 127 dB, respectively. We have added this information to the text. Comment 21: What is the reason to

measure the pressure inside in line 176? Response 21: A pressure sensor inside the glass sphere is used as a negative pressure monitor for determining if there is a leak in the glass sphere. Comment 22: Does "set" mean "group" or "pack"? Better to change the word in line 178. Response 22: "Set" has been corrected to "pack" in the revised manuscript. Comment 23: the landing position or the true position at the seafloor in line 202. Response 23: "True position" has been corrected to "landing position" in the revised manuscript. Comment 24: "fast Fourier transfer(FFT)" should be corrected as "Fast Fourier Transform (FFT)" in line 217. Response 24: "Fast Fourier transfer (FFT)" has been corrected to "Fast Fourier Transform (FFT)" in the revised manuscript. Comment 25: "Fig.8" should be corrected as"Fig.7"; "Fig.9" should be corrected as"Fig.8"; "Fig.10" should be corrected as"Fig.9"; Response 25: The typos have been corrected in the revised manuscript. Comment 26: References should be corrected. Response 26: The references have been corrected in the revised manuscript. Comment 27: The caption of Fig.3 should be corrected. Response 27: The caption of Fig.3 has been corrected in the revised manuscript. Comment 28: Site name of Fig.5 should be simple. Response 28: All of the site names have been renamed in the revised manuscript. Comment 29: Explain "Global Mapper". Response 29: Global Mapper is a mapping software package. Comment 30: At which sampling rate? Or values of sensor (and Amp.) itself in table 1? Response 30: This is at a sampling rate of 2400 Hz for a channel with a sensor, amplifier, and data logger

We would like to reiterate that we are deeply grateful for your comments and suggestions. We have tried our best to improve the manuscript and have made all of the necessary changes in the revised version. We truly appreciate the time and efforts of the editors and reviewers, and we sincerely hope that our corrections will be met with your approval.

Sincerely, Kai Chen

China University of Geosciences Beijing 100083 ck@cugb.edu.cn

---

## Author Response (AR1)

**A Compact Ocean Bottom Electromagnetic Receiver and Seismometer (ID: gi-2019-25)**
**Response to Reviewers**

Dear Editor,

Thank you very much for providing us with the opportunity to revise our manuscript. We appreciate the positive and constructive comments and suggestions from the editor and reviewers for our manuscript entitled "A Compact Ocean Bottom Electromagnetic Receiver and Seismometer" (ID: gi-2019-25). The comments were valuable and helpful for revising and improving our manuscript, as well as for conveying the significance of our research. We carefully examined all comments and made the necessary corrections, which we hope meet your approval. All changes have been made using red text in the manuscript.

The main corrections made to the manuscript and the responses to reviewer comments are summarized below.

*AR1*

**Comment #1:** Thus, I don't really understand how to join the OBEM and OBS data to investigate gas hydrate or petroleum exploration within a few kilometers below seafloor? I would recommend the authors to distinguish what is the scientific purpose of the instrument?

**Response#1:** Marine controlled-source electromagnetic (CSEM) sounding is a new tool available to geophysicists for offshore gas hydrate exploration (Weitemeyer, 2011). This technique has been developed for the detection of deep hydrocarbon reservoirs (Fanavoll, 2010). The OBEM is the receiver that measures the EM field for the marine CSEM and/or MT method. OBS mainly provides deep geological information and is also used to map shallow gas hydrates (Mienert, 2005). Therefore, these two offshore active/passive geophysical exploration instruments could jointly provide a complementary image to identify natural resources and/or geological structures. Thus, joining the OBEM and OBS data acquisition to investigate gas hydrate or petroleum exploration several kilometers below the seafloor is possible.

**Change #1:** We have added a special paragraph from Lines 66 to 73 and appropriate references.

**Comment #2:** How to avoid the seismometer generates noise for magnetic sensors?

**Response #2:** Three 8 Hz omni-directional geophones were used as a seismometer to record artificial earthquake signals. The moving coil geophone may generate EM noise for magnetic sensors, but the electronics (data acquisition circuit board, battery, and geophone) are all shield by ferrite film. The distance between the induction coil and geophone is too large to measure the EM noise. We confirm the EM noise of the geophone test in the magnetic shield room.

**Change #2:** "Ferrite sheets, with a 0.01-mm-thick film on one side and a 0.02-mm-thick adhesive tape on the other, were glued inside the shielding box. These ferrite sheets function primarily as suppressors, blocking EM noise at lower

frequencies and absorbing it at higher frequencies." Updated in the caption of Fig. 1.

Comment #3: Please comparing and demonstrating the accuracy between the USBL attached to the OBEMS and other OBEM.

Response #3: The resulting reduction in the positioning uncertainty leads to significant improvements in the target sensitivity. Acoustic ultra-short baseline communication (USBL) is used to establish the exact receiver positions. The OBEMS integrated the USBL transponder, which is from the Sonardyne GyroUSBL underwater acoustic positioning solution, with an accuracy of approximately 1.5 ‰ of the slant distance. While the slant distance is 2,000 m, we estimate that the receiver positions obtained in this manner are accurate to ~3 m. The OBEM, which is from the EMGS position, is monitored by an acoustic USBL transponders. The OBEM, which is from the SIO accurate navigational data, were collected using a short baseline (SBL) acoustic navigation system. Receiver positions obtained in this manner are accurate to ~3-5 m. The USBL is more conveniently installed and operated than the SBL, characterized by sufficient accuracy.

Change #3: Several explanations have updated from Lines 147 to 156.

Comment #4: I can't find the related descriptions of figures 2 and 4 in the context. Please either add the descriptions or remove these figures.

Response #4: The related descriptions of Figs. 2 and 4 have been added to the manuscript. Figure 2 show the photo of the OBEMS while floating to the surface. Figure 4 shows the data logger installed in the glass sphere.

Change #4: Descriptions of Figs. 2 and 4 have been updated on Lines 120 and 208.

Comment #5: P5, L159: How about the gain of the magnetic sensors?

Response #5: The gain of the magnetic sensor is 300 mV/nT and the output range is ± 5 V.

Change #5: Description of the induction coil has been updated on Line 79.

Comment #6: P6, L206: Which method? Please cite the reference or specify it in detail.

Response #6: A reference has been added (Egbert, G. D.: Robust multiple-station magnetotelluric data processing, Geophys. J. Int., 130, 475– 496, 1999).

Change #6: A reference has been added. (Egbert, G. D.: Robust multiple-station magnetotelluric data processing, Geophys. J. Int., 130, 475– 496, 1999).

Comment #7: P6, L209: At high frequency ranges, the seafloor responses. Please rewrite it.

Response #7: At high frequencies, we observe the sea floor response for both modes as an asymptote to 1 $\Omega$m.

Change #7: "At high frequencies, we observed the sea floor response for both modes asymptotic to 1 $\Omega$m" has updated on Lines 231.

Comment #8: P6, L198: Figure 7 should be replaced by figure 5? P6, L218: Fig.8

should be replaced by Fig. 7? P6, L220: Figure 9 should be replaced by Figure 8? P6, L224: Fig.10 should be replaced by Figure 9?

Response #8: We are sorry for this carelessness. Figure 7 has been replaced by Figure 5. Figure 8 has been replaced by Figure 7. Figure 9 has been replaced by Figure 8. Figure 10 has been replaced by Figure 9.

**Change #8:** Mistakes have been corrected from Lines 243 to 249.

**Comment #9:** Table 1 should specify the seismometer in detailed.

**Response #9:** The sensor type, dynamic range, and gain preamplifier of the seismometer have been added to Table 1.

**Change #9:** The sensor type, dynamic range, and gain preamplifier of the seismometer have been added to Table 1.

**Comment #10:** The figure 4 shows a 70.8 km long high-resolution seismic reflection profile JM00-026 crossing both the Storegga (Mienert et al., 2005) instead of OBS data! As I mentioned the critical key point is that the OBS data can't provide a high-resolution profile as seismic reflection profile. Otherwise, could the authors demonstrate how OBS provides a BSR indicator for gas hydrate? Seismic reflection data were acquired in 2000 onboard R/V Jan Mayen, using a double sleeve gun (0.6 l each) towed at w4 m depth and a floating single-channel streamer at short offsets. The frequency content ranges from 30 to 450 Hz, with peak frequency centered around 100 Hz. The overall quality of the records is high, with good signal to noise ratio and a sub-surface penetration of up to 1 s. Seismic profile JM00-026 presented in this paper (Fig. 4) is 71 km long, with an average shot spacing of 27.2 m. Data processing included frequency filtering, amplitude corrections and Stolt migration (Mienert et al., 2005).

**Response #10:** Due to the small number and sparseness of the OBS, as well as the limited extent of the shot patch, it would be unrealistic to expect an image of comparable quality to the towed-streamer image. However, OBS is feasible and may be a powerful technology for deep water imaging projects. The superior quality observed in the OBS data can be attributed to the following factors (Manley et al., 2005):

1) Geophone deployment in a quiet, seafloor environment. Allowing good coupling with the seafloor may improve the signal-to-noise rate.

2) Wide-azimuth acquisition, which enables discrimination against overburden features that can potentially scatter energy.

3) High fold. The fold of the PZ data is up to 10 times that of the streamer data, enabling a boost in signal-to-noise.

4) PZ summation, which enables receiver-side water-bottom multiple attenuation.

Besides, the CSEM is a new tool available to geophysicists for offshore gas hydrate exploration. Therefore, the OBEMS can be used for the active EM and seismic method for shallow gas hydrate mapping instead of replacing the towed-streamer method.

References:

Manley, D., Mohammed, S., Robinson, N., and Rowland, T.: Structural interpretation of the deepwater Gunashli Field, facilitated by 4-C OBS seismic data, *The Leading Edge*, 24, 922–926, 10.1190/1.2056396, 2005.

**Change #10:** There was no change required for this comment.

*AR2:*

**Comment 1#:** Each single shot waveform, its spectrum, and the background noise spectrum should be indicated in the figure.

**Response 1#:** Figures 9 and 10 have been updated based on your comment.

**Change #1:** Single shot waveforms, spectra, and background noise spectra have been added to Figures 9 and 10 in the revised manuscript.

**Comment 2#:** Omni direction geophone without the leveling (gimbal).Because of the ocean bottom instrument has some amount of tilt on the seafloor, 3-comp. geophones should not be true UD/H1/H2 output. The tilt may be estimated by cross correlations between channels.

**Response 2#:** Omnidirectional geophones were used because traditional geophones cannot effectively and reliably receive vibration signals on an inclined seabed. In addition, an AHRS was installed to measure the orientation and inclination of the geophone for further data processing.

**Change #2:** Explanations have been updated from Lines 124 to 126.

**Comment 3#:** Problem of the use of the flag is a high level noise source at the seafloor, which is well known in the OBS world.

**Response 3#:** This is an unfortunate deficit in the instrumental design. We will remove the flag in future deployments.

**Change 3#:** There were no changes to the text required for this comment

**Comment 4#:** reference in Line 49 and Line 51

**Response 4#:** This reference has been updated.

**Change 4#:** "QUASAR, 2016" has been corrected to "QUASAR 2019" in the revised manuscript on Lines 45 and 49.

**Comment 5#:** Difference between "towed streamer EM system" and "towed-streamer EM system" from line 86 to line 88.

**Response 5#:** "towed-streamer EM system" has been corrected to "towed streamer EM system."

**Change 5#:** "towed-streamer EM system" has been corrected to "towed streamer EM system" in the revised manuscript on Lines 90 and 91.

**Comment 6#:** First appearance of the "OBEMS", in line 101.

**Response 6#:** "Ocean Bottom Electromagnetic Receiver and Seismometer (OBEMS)"

has been added to Line 104.

**Change 6#:** "Ocean Bottom Electromagnetic Receiver and Seismometer (OBEMS)" has been added to Line 104.

**Comment 7#:** Indicate the model name/number and company of the omni-directional geophone in line 119.

**Response 7#:** We have added this information to the revised manuscript. The instrument was an ODG8 geophone manufactured by the Chongqing Geological Instrument Factory.

**Change 7#:** Please see Lines 123 to 125.

**Comment 8#:** "Li-ion batteries" in line 121 should be corrected as "Li-ion rechargeable batteries"

**Response 8#:** "Li-ion batteries" has been corrected to "Li-ion rechargeable batteries" on Line 132.

**Change 8#:** "Li-ion batteries" has been corrected to "Li-ion rechargeable batteries" on Line 132.

**Comment 9#:** Please add the condition of this power consumption clearly in line 124.

**Response 9#:** The power consumption was approximately 1 W at a maximum sampling rate of 2,400 Hz and the power supply module supported data acquisition for ~30 days.

**Change 9#:** "The power consumption is approximately 1 W at a maximum sampling rate of 2,400 Hz and the power supply module supports data acquisition for ~30 days." Updated on Lines 133 to 135.

**Comment 10#:** No explanation about the "U-profile".  Please add some text here or in Fig.1 in line 144.

**Response 10#:** A single U-shaped stainless steel strap connects the two release mechanisms, passing through two stainless steel loops set into the anchors.

**Change 10#:** An explanation of this "U-shape" has been added to Lines 163 and 164 and Fig. 1 in the revised manuscript.

**Comment 11#:** MCXO should be explain as "microprocessor controlled crystal oscillator" in line 147.

**Response 11#:** MCXO has been defined as a "microprocessor controlled crystal oscillator" on Line 166.

**Change 11#:** MCXO has been defined as a "microprocessor controlled crystal oscillator" on Line 166.

**Comment 12#:** This buoyancy looks large, please add the reason why it is required in line 151.

**Response 12#:** The redundant buoyancy was designed for the addition of more batteries for longer seafloor working times. We have added this explanation to the

text.

**Change 12#:** Please see Lines 169 and 170.

**Comment 13#:** "24-bit ADC" should be corrected as "a 24-bit ADC" in line 158.

**Response 13#:** "24-bit ADC" has been corrected to "a 24-bit ADC" in the revised manuscript.

**Change 13#:** "24-bit ADC" has been corrected to "a 24-bit ADC" on Line 176.

**Comment 14#:** Chopper should be an OP-amp, please indicate model number and company from line 158 to line 159.

**Response 14#:** The pre-amplifier for the E-field channel is an ultra-low noise chopper amplifier that has been upgraded from Constable (2013).

**Change 14#:** We have added this information on Lines 176 and 177.

**Comment 15#:** "ADS1282" should be corrected as "a ADS1282 (TI)" in line 162.

**Response 15#:** "ADS1282" has been corrected to "ADS1282 (TI)" in the revised manuscript.

**Change 15#:** "ADS1282" has been corrected to "ADS1282 (TI)" on Line 181.

**Comment 16#:** This value is in your OBEMS system or the ADC only?  Please indicate clearly in line 165.

**Response 16#:** This is for ADC only. We have added this information to the text.

**Change 16#:** "which provides a dynamic range of 130 dB at a 250 Hz sampling rate and a total harmonic distortion (THD) of –122 dB." Please see Lines 183 and 184.

**Comment 17#:** Indicate model and company of the MCU in line 166.

**Response 17#:** This is an ATmega16 from ATMEL.

**Change 17#:** "(MCU) A (AT91SAM9G45 from ATMEL)" and "MCU B (ATmega16 from ATMEL)," have been updated on Lines 186 and 186.

**Comment 18#:** Indicate model and company of the CPLD in line 169.

**Response 18#:** This is an EPM570 from INTEL.

**Change 18#:** "(CPLD) (EPM570 from INTEL)" has been updated on Line 190.

**Comment 19#:** Indicate the model of the MCXO inline 171.

**Response 19#:** This is an MX-503 from Vectron.

**Change 19#:** The MCXO (MX-503 from Vectron) has been updated on Line 192.

**Comment 20#:** The resolution in each sampling rate should be different, please indicate from line 173 to line 174.

**Response 20#:** The sampling rate can be set to 2,400, 600, or 150 Hz, where the dynamic range reaches approximately 115, 121, and 127 dB, respectively. We have added this information to the text.

**Change 20#:** "The selectable sampling rate can be set to 2,400, 600, or 150 Hz, with a dynamic range that reaches approximately 115, 121, and 127 dB, respectively" on

Lines 194 to 195.

**Comment 21#:** What is the reason to measure the pressure inside in line 176?
**Response 21#:** A pressure sensor inside the glass sphere was used as a negative pressure monitor to determine if there is a leak in the glass sphere.
**Change 21#:** Please see Lines 197 and 198.

**Comment 22#:** Does "set" mean "group" or "pack"? Better to change the word in line 178.
**Response 22#:** "Set" has been corrected to "pack" in the revised manuscript.
**Change 22#:** "Li-ion rechargeable battery pack" on Line 200.

**Comment 23#:** the landing position or the true position at the seafloor in line 202.
**Response 23#:** "True position" has been corrected to "landing position" in the revised manuscript.
**Change 23#:** "its landing position" on Line 222.

**Comment 24#:** "fast Fourier transfer (FFT)" should be corrected as "Fast Fourier Transform (FFT)" in line 217.
**Response 24#:** "Fast Fourier transfer (FFT)" has been corrected to "Fast Fourier Transform (FFT)" in the revised manuscript.
**Change 24#:** "Fast Fourier Transform (FFT)" on Line 237.

**Comment 25#:** "Fig.8" should be corrected as"Fig.7";
"Fig.9" should be corrected as"Fig.8";
"Fig.10" should be corrected as"Fig.9";
**Response 25#:** These mistakes have been corrected in the revised manuscript.
**Change 25#:** These mistakes have been corrected from Lines 240 to 246.

**Comment 26#:** References should be corrected.
**Response 26#:** The references have been corrected in the revised manuscript.
**Change 26#:** The references have been corrected in the revised manuscript.

**Comment 27#:** The caption of Fig.3 should be corrected.
**Response 27#:** The caption for Fig. 3 has been corrected in the revised manuscript.
**Change 27#:** The caption for Fig. 3 has been corrected in the revised manuscript.

**Comment 28#:** Site name of Fig.5 should be simple.
**Response 28#:** All of the site names in Fig.5 have been renamed in the revised manuscript.
**Change 28#:** All of the site names in Fig. 5 have been renamed in the revised manuscript.

**Comment 29#:** Explain "Global Mapper".

**Response 29#:** Global Mapper is a mapping software package.
**Change 29#:** "The map was generated using the Global Mapper software package" in the caption of Fig. 5.

**Comment 30#:** At which sampling rate? Or values of sensor (and Amp.) itself in table 1?
**Response 30#:** This is at a sampling rate of 2,400 Hz for a channel with a sensor, amplifier, and data logger.
**Change 30#:** "Channel –3 dB band width @ fs = 2,400 Hz" in Table 1.

We look forward to hearing from you regarding our submission. We would be happy to respond to any further questions and comments that you may have.

Sincerely,
Kai Chen
China University of Geosciences Beijing 100083
ck@cugb.edu.cn